# An Improved Distributed Sampling PPO Algorithm Based on Beta Policy for Continuous Global Path Planning Scheme

**DOI:** 10.3390/s23136101

**Published:** 2023-07-02

**Authors:** Qianhao Xiao, Li Jiang, Manman Wang, Xin Zhang

**Affiliations:** 1School of Electronic Engineering, XI’AN University of Posts&Telecommunications, Xi’an 710121, China; xiaoqianhao@stu.xupt.edu.cn (Q.X.); jiangli@xupt.edu.cn (L.J.); 2School of Physical Science and Technology, Tiangong University, Tianjin 300387, China; 2031121198@tiangong.edu.cn

**Keywords:** artificial intelligence, path planning, reinforcement learning, deep learning

## Abstract

Traditional path planning is mainly utilized for path planning in discrete action space, which results in incomplete ship navigation power propulsion strategies during the path search process. Moreover, reinforcement learning experiences low success rates due to its unbalanced sample collection and unreasonable design of reward function. In this paper, an environment framework is designed, which is constructed using the Box2D physics engine and employs a reward function, with the distance between the agent and arrival point as the main, and the potential field superimposed by boundary control, obstacles, and arrival point as the supplement. We also employ the state-of-the-art PPO (Proximal Policy Optimization) algorithm as a baseline for global path planning to address the issue of incomplete ship navigation power propulsion strategy. Additionally, a Beta policy-based distributed sample collection PPO algorithm is proposed to overcome the problem of unbalanced sample collection in path planning by dividing sub-regions to achieve distributed sample collection. The experimental results show the following: (1) The distributed sample collection training policy exhibits stronger robustness in the PPO algorithm; (2) The introduced Beta policy for action sampling results in a higher path planning success rate and reward accumulation than the Gaussian policy at the same training time; (3) When planning a path of the same length, the proposed Beta policy-based distributed sample collection PPO algorithm generates a smoother path than traditional path planning algorithms, such as A*, IDA*, and Dijkstra.

## 1. Introduction

Trade transportation is the basis of economic development, and, with the growth in trade volume in various countries, shipping accounts for more than 2/3 of international cargo transportation. Accidents, however, have been continuously posing risks to maritime transportation; according to one survey, more than 80% of accidents are caused by human factors in the process of ship navigation [1]. Path planning, as one of the key technologies for traditional autonomic navigation scheme, aims to find a safe and economic path for ships and has received increasing attention from both industry and academia in recent years [2].

The plan of the path provides the optimal route based on various factors such as distance, time, etc. Considering divergent factors, path planning can be divided into local path planning and global path planning [3]. Local path planning indicates that the changes in the surrounding environment are unknown (mainly reefs and dynamic obstacles), and the path needs to be corrected based on sensor data in real time. Global path planning involves planning a route from the starting point to the endpoint, founded on known environmental information at a large geographic scale (1 km to 100 km level, or more).

This paper presents an in-depth study of global path planning. Traditional path planning algorithms currently used to solve global path planning include the A*, Dijkstra, and artificial potential energy field methods. Most of these algorithms use spatial discretization to represent the environment as equal-sized raster maps and discretize the action space into four-connected or eight-connected [4]. The limitations of these path planning algorithms include the following. First, these algorithms rely on prior knowledge of the system when used for global path planning. Consequently, if there are changes in the environment, such as the emergence of new obstacles, it is necessary to obtain updated and accurate information about the environment in order to re-plan the route. In addition, these algorithms do not take into account the ship’s navigational capability, only searching the driving path without giving the bow heading and velocity vector of the nodes in the path. Moreover, these algorithms search the way topologically, leading to much higher computation and lower efficiency required for planning with the increasing complexity in the planning scene.

To address the challenges, as mentioned before, of traditional path planning, Sun proposed a dynamic trajectory generation model for ship rust removal robots based on a bionic neural network and utilized a kinematic state matching search method to achieve dynamic path planning while considering the robot’s kinematic model [5]. However, during actual testing, it was observed that the proposed method lacked robustness in the planning process, making it unable to guarantee successful planning. To improve the robustness and stability of the algorithm in the path planning task, Shen et al. from Dalian Maritime University proposed an intelligent collision avoidance navigation method for unmanned ships, which considered navigation experience and collision avoidance rules using Deep Competitive Q-learning (DCQN) with the A* algorithm [6]. Similarly, Wang et al., also from Dalian Maritime University, proposed the DRLOAD algorithm based on the Deep Q-Network (DQN), which achieved a multi-dimensional navigation by designing a reward function specifically for safe obstacle avoidance [7]. Both methods employed value learning methods in reinforcement learning to solve path planning problems in a discrete action space. Nevertheless, these approaches faced issues with the reward function’s design and the ship propulsion strategy’s incomplete structure during the planning process. Another relevant contribution is the path planning algorithm proposed by Kuczkowski, which is based on the Deep Deterministic Policy Gradient (DDPG) algorithm. This approach integrates several subgoals of path planning, collision avoidance, compliance with the Convention on the International Regulations for Preventing Collisions at Sea (COLREGs), and task completion by designing four reward functions [8]. Kuczkowski successfully achieved path planning and dynamic obstacle avoidance that satisfied COLREGs in complex scenarios. While adequately designing the reward functions and providing each path node’s coordinates and velocity vectors during planning, the environment modeling did not consider the ship’s kinematic model.

A summary of the above related work is presented in Table 1.

Based on the above research work, this paper proposes a distributed sampler PPO algorithm based on Beta policy to address the pain points, i.e., that traditional path planning increases computationally in the case of rising scenario complexity and cannot directly deal with high-dimensional data. Furthermore, it addresses the problems that reinforcement learning algorithms face in solving path planning problems, such as the unreasonable design of the reward function—most algorithms only perform path planning in the discrete action space and primarily use deterministic policy in planning. The main technical contributions of this paper are as follows:Designing and constructing a simulation environment that conforms to the ship propulsion characteristics, using the Box2D physics engine as the framework;Constructing a reward function that integrates the mainline reward and auxiliary reward, where the mainline reward is determined based on the distance between the agent and the arrival point, while the auxiliary reward is potential-based, superimposed from boundary control, obstacles, and arrival point;Proposing a distributed sampling policy on the basis of the stochastic policy algorithm PPO, which significantly enhances sampling performance and enables robust ship path planning in continuous domain spaces;Introducing the Beta policy as a replacement for the Gaussian policy in the action selection policy. This substitution effectively resolves the issue of action boundary constraints.

The advantages of the algorithm proposed in this paper are reflected as follows. By employing the stochastic policy algorithm, this paper achieves better planning results by simulating a ship’s propulsion characteristics in a continuous state and action space. The integration of the mainline and auxiliary rewards results in a smooth trajectory, with each planned path node providing position and velocity vectors. Our proposed algorithm attains shorter planning paths than other reinforcement learning algorithms. Moreover, when compared to traditional path planning methods with similar path lengths, our algorithm generates smoother trajectories.

## 2. Background

In solving global path planning problems by reinforcement learning, the learning tasks chosen vary for the different discrete and continuous action spaces in global path planning. Reinforcement learning is usually described by a Markov decision process (MDP), which consists of a state-space S and an action space A. In MDP, the individual who performs the learning and decision-making is called the agent, and the range with which the agent is interacting is called the environment. At the discrete moment t, the agent observes the states $S_{t}\left(S_{t}\in S\right)$ fed by the environment and gives feedback actions $A_{t}\left(A_{t}\in A\right)$ to all these states, according to the policy π, where the policy can be deterministic or stochastic. The environment provides a reward $R_{t}$ in response to the action and moves to the next state $S_{t+1}\in S$; the state–action pairs generated by this agent-environment interaction constitute a trajectory:$$S_{0},A_{0},R_{1},S_{1},A_{1},R_{2},S_{2},A_{2},R_{2},\ldots S_{n},A_{n},R_{n+1}$$

The learning tasks of reinforcement learning (RL) can be divided into value-based and policy-based categories.

### 2.1. Value-Based Methods for Discrete Domain

Traditional RL uses the Q-learning method and stores the transfer state by Q-table when applied to path-planning tasks. To solve the problem that, when the state complex increases, the efficiency of the Q-learning update drops, DQN, as proposed in 2013, adds the deep neural network and experience replay based on Q-learning [9]. Although it improves the policy performance dramatically and can be used to handle high-dimensional state-space tasks, DQN generates an unavoidable overestimation problem in the process of action-value Q estimation, which occurs because DQN performs action-value estimation based on the Bellman Equation (Equation 1):(1)$$Q^{*}\left(s,a\right)=E_{\pi }\left[r_{t+1}+\gamma \cdot \max_{a\in \theta^{\prime }}Q^{*}\left(s_{t+1},a^{\prime };\theta_{t}^{\prime }\right)\right]$$

In this context, the action-value function $Q^{*}\left(s_{t},a_{t}\right)$ is defined as the expected value of the immediate reward $r_{t+1}$ obtained by taking action $a_{t}$ in the current state $s_{t}$, plus the expected value with discounted maximum action-value $Q^{*}\left(s_{t+1},a^{\prime };\theta_{t}^{\prime }\right)$ of the next state $s_{t+1}$, corresponding to the action $a^{\prime }$ chosen by the target policy π with parameters $\theta_{t}^{\prime }$.

Here, $\theta_{t}^{\prime }$ represents the target network parameters for the policy π, and γ is the discount factor 0≤γ<1. The target network always selects the action that maximizes the action-value $Q^{*}\left(s_{t},a_{t}\right)$ according to its own policy $\theta_{t}^{\prime }$, which leads to bootstrap. Silver et al. [10] replaced the bootstrap action generated by the target network with online network policy to update the action value estimation:(2)$$Q^{*}\left(s,a\right)=E_{\pi }\left[r_{t+1}+\gamma \cdot Q\left(s_{t+1},{\mathrm{argmax}}_{a\in \theta }Q\left(s_{t+1},a^{\prime };\theta_{t}\right);\theta_{t}^{\prime }\right)\right]$$

The online network $Q\left(s_{t+1},a;\theta_{t}\right)$ is used for action selection, and the corresponding action is used for target calculation; due to a delay in the update of the network target, the actions that correspond to the value q-max of the online network are not necessarily identical to the actions that correspond to the value q-max in the network target. Thus, the problem of overestimating is reduced.

### 2.2. Policy-Based Method for Continuous Domain

This value-based learning method cannot solve the stochastic policy problem well because it is a deterministic policy in essence. The policy gradient-based [11] reinforcement learning algorithm optimizes a policy function by fitting a continuous function representing the current task performance metric and using gradient ascent. In this state, the value function can be used to assist the policy training, without need for action choice. The policy gradient-based approach has more robust convergence than the action-value-based approach; the policy gradient $\nabla_{\theta }J\left(\pi_{\theta }\right)$ is expressed as follows:(3)$$\begin{array}{cc} \nabla_{\theta }J\left(\pi_{\theta }\right) & =\int_{S}\rho^{\pi }\left(s\right)\int_{A}\nabla_{\theta }\pi_{\theta }\left(a|s\right)Q^{\pi }\left(s,a\right)dads\\ \; & =\int_{S}\rho^{\pi }\left(s\right)\int_{A}\pi_{\theta }\left(a|s\right)g_{q}dads\\ \; & =E_{s∼\rho^{\pi },a∼\pi_{\theta }}\left[g_{q}\right] \end{array}$$
where $\rho^{\pi }\left(s\right)$ is the state distribution, $\pi_{\theta }\left(a|s\right)$ denotes current state s according to the policy sampled by the parameter θ, and $g_{q}$ represents the policy gradient estimator with $Q_{\pi }\left(s,a\right)$ as the target:(4)$$g_{q}=\nabla_{\theta }\log\pi_{\theta }\left(a|s\right)Q^{\pi }\left(s,a\right)$$

In the actual calculation operation, the policy gradient is estimated by sampling $g_{q}$ in large numbers, so the sample mean of $g_{q}$ converges to its expected value $E\left[g_{q}\right]=\nabla_{\theta }J\left(\pi_{\theta }\right)$ when the number of samples is sufficient.

However, the policy gradient based method has the problems of falling into local optima and inefficient policy evaluation. To solve these problems, Sutton et al. proposed Actor–Critic. The Actor refers to the policy function, i.e., the agent action decision part. Critic refers to the value function that can be regarded as an evaluator of the current state. The Critic and Actor parameters are updated by the data obtained from the real-time interaction between the agent and the environment, which makes the algorithm converge.

Schulman et al. proposed the Proximal Policy Optimization (PPO) algorithm [12] by using the Trust Region Policy Optimization (TRPO) [13] surrogate objective; the PPO clips the surrogate objective to a smaller range and replaces the advantage function $A^{\pi }\left(s,a\right)$ in the agent with a Generalized Advantage Estimator (GAE) [14], which is used for high-dimensional continuous action training by GAE to obtain stable policy. To ensure continuous exploration, even in states where the policy is relatively mature, PPO incorporates entropy rewards when calculating the surrogate objective. Additionally, the inherent advantage of its stochastic policy in path exploration prevents the algorithm from ending up trapped in local optima during the search process [15].

This paper uses PPO as the baseline algorithm for global path planning implementation and optimization. Firstly, we conducted path planning environment design, and in the environment design process, we needed to consider the state-space and action space structure for deep reinforcement learning. For the algorithm to converge properly, we needed to design the reward function to be highly matched for the state-space of the global path planning task. The rationality of the reward function design directly guides the algorithm’s convergence speed and policy performance. The PPO limits the policy update magnitude by clipping the surrogate objective, so that the old and new policies do not produce significant changes in the update and, thus, obtain more stable policy updates.

## 3. Interaction Environment Design

Deep reinforcement learning updates the state through constant interaction between the agent and the environment, and optimizes the policy to maximize the reward. This method requires environment design, including state-space, action space, and reward function. The environment design determines the scene structure and interaction logic and adds constraints to each scene element.

The simulation environment is constructed using the Box2D physics engine for structural development, and the Pyglet module is utilized for rendering the plane frame. The size of the plane frame structure is set as n∗n, with a render scale of *s*. Obstacles are generated at the center of the frame, using a generation scale of τ. The obstacle area is divided into a grid space of k∗k, with a grid radius of *a*. The obstacle radius ranges from 0.2a to 1.5a, and the number of obstacles generated falls within the range $\left[k,k^{2}-k\right]$. To prevent overlapping, an offset is applied to the obstacle generation coordinates. This offset follows a uniform distribution $U\left(-n*\tau *0.05,n*\tau *0.05\right)$, ensuring that no obstacles overlap with each other. Unlike other path planning simulation scenarios for RL, this simulation environment increases the ship and the endpoint generation range to improve the algorithm’s robustness. As shown in Figure 1, the orange-covered area is the hull generation area, and the blue area is the arrival point generation area.

### 3.1. State-Space Design

The state information used for the global path planning task of deep reinforcement learning can be divided into the sensor data sequence and the image from the environment. DRL extracts the relationship between the features and the policy from training. Due to the sensor data sequence’s low dimension, it is easy for the agent to acquire the policy according to the shift features; the image contains a higher dimensionality message, so the agent needs to learn the policy from the image extracted feature. As the network processes the state information differently, we designed two state-spaces, described below.

#### 3.1.1. Environmental Feedback Sensor Data Sequence-Based State-Space

To make the agent perceive the feature shifts in the motion, we designed an approach using LIDAR to measure the distance to avoid obstacles, as shown in Figure 2. We set the ship’s radius as *b*; the effective reflection range of LIDAR is within *h* times the vessel’s radius, expressed as $\left[b,h*b\right]$. We set the horizontal field angle of the detection rays to ρ and uniformly arranged the *l* ray beams within the field. During the ship’s motion, if another object truncates the ray, it is considered that the object is closing. In this case, we gave the ray identifier $flag_{i}$ associated with the intercepted object, as shown in the following equation:(5)$$flag_{i}=\left\{\begin{array}{cc} -1, & \mathrm{object}\;\mathrm{dected}\\ 0, & \mathrm{otherwise}\\ 1, & \mathrm{current}\;\mathrm{goal} \end{array}\right.$$

Calculate the distance to the truncation point, denoted as $ray_{i}$; the vector is shown in Equation:(6)$$G=\left\{\left(flag_{1},ray_{1}\right),\left(flag_{2},ray_{2}\right),\ldots ,\left(flag_{l},ray_{l}\right)\right\}$$

The *l*-beam returns detection feedback based on the environmental information sensed by the vessel’s current position. To reduce the computational effort of the algorithm, we find when l=24 can obtain the best efficiency.

The data sequence state-space consists of five different sensor states:Position sensor: obtain the current vessel’s coordinates $\left(x_{i},y_{i}\right)$;Speed sensor: obtain the current vessel speed vector $\left(v_{x},v_{y}\right)$;Heading sensor: obtain the current ship’s bow direction (true heading TC) $\theta_{N}$;Radar: obtain the perception information about the vessel $G=\left[g_{1},g_{2},\ldots g_{l}\right]$;Distance sensor: obtain the straight-line distance *D* to the endpoint.

Having completed the state-space design, we process the data and send it to the network for inference. During processing, the data are normalized, stacked, and formed into consecutive frames by superposition. The normalization, which, according to the maximal value of different sensors, is present to reduce the scale difference between samples and make the gradient update more stable, thus helping faster convergence to the optimal policy.

Before executing the continuous frame data overlay, we convert the data dimension to $\left[1,frame_{overlay},state_{len}\right]$, concatenate along the direction of dim = 1, and flatten the concatenated data at the last dimension, i.e., $\left[1,frame_{overlay}*state_{len}\right]$. By overlaying multiple data frames, the agent can understand the current motion direction and speed of the ship and, thus, make reasonable decisions. In contrast, it is not available in single-frame data.

#### 3.1.2. Image-Based State-Space

The image-based state-space contains the current agent scenario image and navigation data. The agent uses the image to evaluate the relationship between the ship and the arrival point, boundary, and obstacles to realize the perception of the surrounding area. During the data processing, we divide the image and navigation information into two parts, the image part (which handles grayscale and normalization), and the navigation part (which normalizes). After that, we stack features to the same dimension at each part, and the network fuses the features of these two parts.

In image processing, the image shape which returned from Pyglet rendering RGB data is $\left[3,sn,sn\right]$. We convert the three-channel RGB image into a single-channel grayscale map, and the image size is appropriately reduced to $\left[1,1,x,x\right]\;\left(x<sn\right)$ for policymaking, while the grayscale image is normalized in the image processing.

The image-based state-space consists of the following vectors:Current scenario seascape: image data in the shape of $\left[1,1,x,x\right]$ with normalization operation;Position vector: spatial coordinate information $\left(x_{i},y_{i}\right)$ where the current ship is located;Velocity vector: obtain the current ship’s velocity vector information $\left(v_{x},v_{y}\right)$.

We keep the same data overlay operation with the data sequence state-space and concatenate the two groups of processed image and navigation feature data along the direction of dim = 1 (the image shape is $\left[1,1,x,x\right]$ and the navigation data shape is $\left[1,1,4\right]$). Finally, we obtain the overlayed feature data with the image shape $\left[1,frame_{overlay},x,x\right]$ and the navigation data shape $\left[1,frame_{overlay}*4\right]$.

### 3.2. Action Space Design

The Box2D physics engine is used in the environmental design for ship action control. The simulated trajectory of the ship’s rigid body is calculated by the semi-implicit Euler method to generate a planning curve in path planning that matches its physical characteristics for simulated propulsion control. To ensure accurate planning paths, the action space is designed in two dimensions, i.e., angle and propulsion power:Angle dimension: In this path planning model, the angle design uses continuous space, ranging from 0 to 360 degrees, and is mapped normalized to the interval $\left[0,1\right]$; differently from traditional discrete angle design (such as four-connected or eight-connected actions), the continuous angle control allows the algorithm policy to perform more accurate navigation control to achieve more precise planning;Propulsion power dimension: Propulsion power is the power generated by the ship’s thrusters. In this model, we use the Box2D to complete the propulsion power design, which considers the effects of thrust derating and fluid resistance. The maximum value of the main propulsion power is set to 70. Consistent with the angle treatment, the propulsion power is also normalized and mapped to the interval $\left[0,1\right]$.

In summary, the action space is designed with continuous control in angle and propulsion power dimensions. The agent can flexibly adjust the thrust angle and power to competently adapt to the physical characteristics of the ship control in the path planning task and, thus, perform accurate planning in different scenarios.

### 3.3. Reward Function Design

The reward function should correlate highly with the state-space to make the agent learn the expected policy [16]. We divided the reward function into mainline and auxiliary reward, where the design of mainline reward $R_{dist}$ mainly considers the distance between the ship and the destination, and the creation of auxiliary focuses on the effect of arrival point, boundary control, and obstacles on the traveling state. We set the initial distance between the vessel and the target generation position at initializing stage as $dist_{init}$. While the agent interacts with the environment, we record the distance between the ship and the endpoint as $dist_{rec}$. With the hull position changing, if the current distance $dist_{cur}$ is less than $dist_{rec}$, a positive reward is given and rewritten $dist_{rec}$ by $dist_{cur}$; otherwise, it is returned as negative. The main line reward is as follows:(7)$$R_{dist}=\left\{\begin{array}{cc} \frac{-dist_{cur}}{dist_{init}}, & \mathrm{if}\;dist_{cur}\geq dist_{rec}\\ 1-\frac{dist_{cur}}{dist_{rec}}, & \mathrm{if}\;dist_{cur}<dist_{rec} \end{array}\right.$$

To prevent the agent from abnormal state-shifting in policy learning, which leads to failure to learn the excepted policy, we solved this problem by creating a reward function with a potential-based reward shaping method [17], i.e., the repulsive field supported by obstacles and boundary control, and the gravitational field supported by arrival points. The agent is given a decreasing penalty when it is close to the endpoint and an increasing penalty when it is close to the obstacles or boundary. This reward format can help the agent learn the robust correlation between state and reward, thus enabling the agent to accelerate the agent policy convergence while maintaining the invariance of the optimal policy.

The reward function is designed to remap the rendered image of n∗n onto a plane of size $m*m\left(n>m\right)$ and calculate the potential field value at each point on that plane. The specific details are as follows:The repulsive field generation is calculated by the potential decline of the 2.5 times $obsc_{R}$ obstacles range. The distance from the ship to the obstacle’s center is $dist_{C}$. We design the reward function satisfy the factors of the potential; note that the bias calculated by $obsc_{R}$ is α. After the matrix is generated, the potential matrix generated for each obstacle is normalized and summed, as shown in Figure 3a and explained below:
(8)$$\begin{array}{cc} R_{obsc} & =\left\{\begin{array}{cc} \ln\left[obsc_{R}*0.05\right]-\alpha , & \frac{dist_{C}}{obsc_{R}}\in \left[0,1\right)\\ \ln[dist_{C}-obsc_{R}*0.95]-\alpha , & \frac{dist_{C}}{obsc_{R}}\in \left[1,2.5\right) \end{array}\right.\\ R_{obsc} & =1-\mathrm{norm}\left(R_{obsc}\right) \end{array}$$The arrival point-based gravitational field is generated by calculating the distance between different positions of the potential matrix and the arrival point, and calculating the gravitational potential decay based on the span. Setting the radius of the arrival point as $reach_{R}$, the distance between the current position and the arrival point is $dist_{C}$. The bias calculated by $reach_{R}$ is β. Figure 3b shows the arrival point-based potential plane, as calculated below:
(9)$$\begin{array}{cc} R_{reach}= & \left\{\begin{array}{cc} \beta -\ln\left[reach_{R}*0.5\right], & dist_{C}\leq reach_{R}\\ \beta -\ln\left[dist_{C}-reach_{R}*0.5\right], & dist_{C}>reach_{R} \end{array}\right.\\ R_{reach}= & 1-\mathrm{norm}\left(R_{reach}\right) \end{array}$$The Repulsive field generation, based on boundary control, makes the agent consider the unknown of the boundary situation in path planning when the agent output action makes the ship move towards the boundary, gradually increasing the penalty. Thus, this ensures the route selected by the agent favors the known and clear center region and improves the planned route’s security. If we set that the penalty boundary is $bound_{limit}$, the distance between the current position and the nearest edge is $dist_{limit}$, and the bias calculated by $dist_{limit}$ is γ. Figure 3c shows the potential plane based on boundary control, calculated as follows:
(10)$$\begin{array}{cc} R_{bound} & =\left\{\begin{array}{cc} \frac{1}{-\ln\left[dist_{limit}+1\right]+\gamma }, & dist_{limit}<bound_{limit}\\ 0, & otherwise \end{array}\right.\\ R_{bound} & =\mathrm{norm}\left(R_{bound}\right)-1 \end{array}$$

From the above analysis, after obtaining the potential matrices with obstacles, arrival points, and boundary control, we can superimpose them and normalize them to obtain the complete potential-based reward function, as shown in Figure 4.

## 4. Beta Policy-Based Distributed Sampling PPO

When applying the policy gradient method for continuous domain control path planning, the DDPG [18] algorithm is usually used to solve complex tasks with high-dimensional data in the continuous domain. However, since DDPG is a deterministic policy, exploration noise needs to be added at the early stage of training to help the agent explore the environment. Suppose the agent cannot obtain enough mainline rewards during exploration. Under the influence of sparse rewards and deterministic policy, DDPG quickly converges to a lousy policy [19], which makes the actor enter a deadlock state, which cannot be jumped out of by increasing mainline rewards later.

DDPG suffers from long training time, low exploration efficiency, and unstable learning strategies in path planning tasks, leading to poor reproducibility and unsuitable path planning in complex environments.

### 4.1. PPO Algorithm Applied to Global Path Planning Task and Optimization

As an on-policy algorithm, PPO uses the surrogate objective to optimize the policy, making it available to figure out discrete and continuous action space. PPO performs action sampling through the latest version of the stochastic policy, which helps the algorithm complete the environment exploration at the early training stage. With training progress, the policy gradually changes from high entropy to low entropy. Hence, the stochastic policy also benefits when facing complex stochastic environments.

#### 4.1.1. PPO Algorithm Operation

The PPO algorithm is a variant of the policy gradient method, which typically calculates a policy gradient estimator and uses the stochastic gradient ascent to optimize. PPO uses the clipping of the surrogate function to reduce the problem of excessive deviation of the new policy from the old one:(11)$$\begin{array}{cc} L\left(s,a,\theta ,\theta_{old}\right)= & \min\left(\frac{\pi_{\theta }\left(a|s\right)}{\pi_{\theta_{old}}\left(a|s\right)}A_{\pi_{\theta_{old}}}\left(s,a\right),\mathrm{clip}\left(\frac{\pi_{\theta }\left(a|s\right)}{\pi_{\theta_{old}}\left(a|s\right)},1-\epsilon ,1+\epsilon \right)A_{\pi_{\theta_{old}}}\left(s,a\right)\right) \end{array}$$

Above, $\pi_{\theta }$ and $\pi_{\theta_{old}}$ denote the new and old policies, and the hyperparameter ϵ is used to limit the difference between the two policies. Suppose, in state *s*, $A_{\pi }\left(s,a\right)>0$; this indicates that the action-value brought by the selected act is greater than the average in the current state. In that case, the update towards the gradient ascent direction increases the selection of actions in such states. $A_{\pi }\left(s,a\right)<0$ indicates the action-value brought by the selected act is lower than the average in the current state, and the update occurs in the gradient descent direction to decrease the selection of actions in such states. With the Actor and Critic training, the Critic evaluates more accurately, and the policy network’s distribution entropy decreases with the training to the convergence state.

#### 4.1.2. Actor–Critic Network Structure

The Actor network uses the same feature extraction structure as the Critic network. However, the feature fusion differs due to the different output dimensions; the Actor and Critic networks vary slightly depending on the state-space.

The network for processing the data sequence state-space is a four-layer fully connected structure, and the hidden layer is [400, 300, 300]; the network for processing the image state-space adds feature extraction layers to extract information from image state, and the structure of the Actor in the image state-space is shown in Figure 5.

The image feature extraction structure consists of four convolutional layers. The first three layers use the same convolutional architecture of DQN when processing Atari images [9]. The fourth layer adds a convolutional layer with an output channel of 256, a kernel of 3∗3, strides of 1, and a padding of 1. The extracted feature is concatenated with the current ship’s navigation information and then fed into the fully connected network with the hidden layer of [1124, 500, 300]. Finally, the mean and variance of Gaussian distribution are output.

Since the current model is used to solve the global path planning task, the model can be made to learn the spatio-temporal relationship of object motion on consecutive time frames through data frame superposition. To match the continuous data frame superposition structure used in the two state-space designs, we consider multi-frame superposition in the design of both model network structures.

### 4.2. Problems of PPO Algorithm in Global Path Planning

When training with PPO for images, the different areas may lead to the environment not being fully explored, i.e., the generation of random positions can cause the agent to enter the termination state early due to collision with the object prematurely. This results in an extreme lack of samples, making it too difficult to train at some starting positions owing to the insufficient positive feedback, resulting in the agent learning only those policies for which data are easily obtained in the training process. Figure 6a shows the test results of six positions selected by equal spacing. From Figure 6a (subgraphs c and d), the agent can learn a better policy at the corresponding starting position in graph a due to the uneven sample acquisition. In contrast, the policies adopted by the agent at the related starting points in the b, c, e, d, and f graphs are similar to those acquired in graph a, indicating that the uneven sample acquisition during the agent’s interaction with the environment affects agent policy learning.

### 4.3. Distributed Sample Extraction PPO

According to the problems encountered in Section 4.2, a distributed sampling strategy based on region partitioning was designed, w hich separates execution from training; each executor explores from different regions and interacts with the environment [20].

The algorithm consists of N sub-processes interacting with the environment and collecting samples into their respective buffers. The process initializes the ship’s position from its sub-regions (Figure 7) and interacts with the environment. After collecting T iter in the buffer, the worker calculates the advantage function and discount reward from the collected data and sends it to the main process in a package. When the main process completes training, the workers pull back the updated parameters from the global and use the new policy to interact.

After the global network collects the data sent back from workers, N∗T samples are shuffled and cut into mini-batches for training. Compared with the A3C algorithm [21], which calculates the gradient by sub-process and uploads the grad to the global, the global network accumulates the grad and optimizes the policy. The newer distributed sample extraction PPO uses sub-processes to collect data and sends them to the main process for optimization. In this sampling method, data from different starting regions can be further integrated, sharing data among workers and increasing sample diversity. The experimental results in Section 5.1 show that this improved method significantly enhanced the robustness of the agent policy, which indicates that the distributed sampling policy based on region division is sufficient to make progress on complex exploration problems.

Figure 8 shows the distributed sampling policy based on region division. On the one hand, the number of samples at each initial position is balanced by region interaction. On the other hand, the distributed sampling policy obtains several times the data of the PPO algorithm over the same duration, thus reducing the convergence time. The robustness of the algorithm is significantly improved due to the rising number of samples. Figure 6b shows the test results. Compared with the PPO baseline (Figure 6a), the distributed sampling policy with area division can achieve better performance in the test of starting positions at equal intervals.

### 4.4. Improvement of Action Policy Based on Beta Distribution in Continuous Action Space

In stochastic policy continuous action control problems, the action sample is typically obtained from a Gaussian distribution $N\left(\mu ,\sigma^{2}\right)$ by predicting the mean and variance in the neural network and optimizing the parameters by computing the gradient of the policy relative to μ and σ. However, the Gaussian distribution is an infinite support distribution. The mean and variance predicted by the neural network are limited by the action space when used in the Gaussian distribution, causing any action sampled from the distribution to be forcibly mapped back into the action space if it exceeds the action boundary. As shown in Figure 9, the Actor calculated the rudder angle according to the specified state and received the distribution parameter of μ=0.7 and σ=0.8, due to the rudder angle being limited to $\left(-1,\;1\right)$. Still, the probability density $\pi_{\theta }\left(a|s\right)$ corresponding to the action a at any position in the Gaussian distribution is greater than 0. To analyze this process, assume that the action space of rudder angle $A=\left[-1,1\right]$, with no clipping in the distribution sampling; then, clipping operates only in the environment. The gradient calculation uses the non-clipping action when computing the surrogate objective in PPO; the policy gradient estimator becomes $g_{q}^{\prime }=\nabla_{\theta }\log\pi_{\theta }\left(a|s\right)A^{\pi }\left(s,a^{\prime }\right)$, where $a^{\prime }$ denotes the clipped action, and $A^{\pi }\left(s,a^{\prime }\right)$ represents the advantage function generated in the clipped action. By calculating the gap between the gradient $E\left[g^{\prime }\right]-\nabla J_{\theta }\left(\pi_{\theta }\right)$, the expected policy gradient is as shown:(12)$$\begin{array}{cc} E\left[g_{q}^{\prime }\right]-\nabla_{\theta }J\left(\pi_{\theta }\right) & =\int_{S}\rho_{\pi }\left(s\right)[\int_{-\infty }^{-1}\pi_{\theta }\left(a|s\right)\nabla_{\theta }\log\pi_{\theta }\left(a|s\right)\left[Q\left(s,a^{\prime }\right)-Q\left(s,a\right)\right]da\\ \; & +\int_{1}^{\infty }\pi_{\theta }\left(a|s\right)\nabla_{\theta }\log\pi_{\theta }\left(a|s\right)\left[Q\left(s,a^{\prime }\right)-Q\left(s,a\right)\right]da]ds \end{array}$$

It could be seen from the formula that the calculation is unbiased within the action space A. This boundary effect caused by the Gaussian distribution produces a deviation in action sampling, which is passed to the calculation of the policy gradient, affecting the training speed.

Chou et al. used the Beta distribution to replace the Gaussian distribution in a continuous control task and trained it faster than the infinite support distributions in the policy gradient algorithm (TRPO, ACER) [22]:(13)$$f\left(x;\alpha ,\beta \right)=\frac{1}{\;B(\alpha ,\beta )}x^{\alpha -1}{\left(1-x\right)}^{\beta -1}$$

$B\left(\alpha ,\beta \right)=\frac{\Gamma \left(\alpha \right)\Gamma \left(\beta \right)}{\Gamma \left(\alpha +\beta \right)}$ represents the Beta function, which ensures that $f\left(x;\alpha ,\beta \right)$ integrates to 1 over the interval $\left[0,1\right]$. The Beta distribution takes values in the range $x\in \left[0,1\right]$, the shape of the distribution is determined by its shape parameters α and β, and, as a finite-support distribution, it does not produce a Gaussian-like boundary effects. Therefore, it is unbiased in the gradient computation and converges faster than other infinite support distributions.

In this paper, to ensure that the action outputs conform to the different limits in the action space, it is necessary to remap the range of the Beta distribution. Therefore, the Beta policy is defined as $\pi_{\theta }^{beta}=f\left(\frac{x+c}{2*c};\alpha ,\beta \right)$. The value of parameter *c* is determined based on the action range in different dimensions of the action space, thereby avoiding the issue of sampling actions beyond the action boundaries.

The Beta distribution is generated by using policy θ to estimate the parameters α and β. Ensuring α,β>1 makes the distribution a concave probability density curve in the interval with a single-peaked shape. In the experiments, when changing the sampling distribution to Beta with the same parameter settings, the training speed was faster than Gaussian, and the accumulated reward was higher than the Gaussian policy for the same training time.
**Algorithm 1** Beta policy-based distributed sampling PPO (***Worker***)1:Set the training episode *E*, maxstep *T*2:Initialize the actor network $\pi_{\theta }$, and critic network $V_{\phi }$ with the random parameters θ, ϕ3:Receive ***event*** flag from main processing4:**for** 
$e\in \left\{1,2,\cdots ,E\right\}$
 **do**5:    According the divided sub-region random generate start point6:    **for** $i\in \left\{1,2,\cdots ,T\right\}$ **do**7:        Run $\pi_{\theta }$ w.r.t Beta Policy, collect $\left\{s_{t},a_{t},r_{t}\right\}$8:        Caculate decayed Reward $\hat{R_{t}^{\left(i\right)}}=\sum_{t=i}^{T-1}r_{t}*\gamma^{T-1-t}+V_{\phi }\left(s_{t}\right)$9:        Caculate $\delta_{t}^{V}=r_{t}+\gamma V_{\phi }\left(s_{t+1}\right)-V_{\phi }\left(s_{t}\right)$10:        Caculate GAE $\hat{A_{t}^{\left(i\right)}}=\sum_{l=0}^{i-1}{\left(\gamma \lambda \right)}^{l}\delta_{t+l}^{V}$11:    **end for**12:    Send *T* length trajectories buffer $g=\left\{s_{t},a_{t},r_{t},\hat{R_{t}^{\left(i\right)}},\hat{A_{t}^{\left(i\right)}}\right\}$ to global through pipe13:    **if** not ***event.is_set()*** **then**14:        ***event.wait()***15:    **end if**16:    Pull Global parameter to local17:**end for**

**Algorithm 2** Beta policy-based distributed sampling PPO (***Global***)
1:Set the training episode *E*, Number of workers *N* and training iterations *K*2:Initialize the actor network $\pi_{\theta }$, and critic network $V_{\phi }$ with the random parameters θ, ϕ3:Initialize ***event*** flag4:**for** 
$e\in \left\{1,2,\cdots ,E\right\}$
 **do**5:    Initialize global buffer *G*6:    ***event.clear***7:    **for** $p\in \left\{1,2,\cdots ,N\right\}$ **do**8:        Receive worker buffer *g* from pipe and collect to global buffer *G*9:    **end for**10:    Sample shuffle, divided N worker samples to K iter batch, K∗batch_size<NT11:    **for** $iter\in \left\{1,2,\cdots ,K\right\}$ **do**12:        Caculate $\pi_{\theta }\left(a_{t}|s_{t}\right)$ using $s_{t}$13:        $J\left(\theta \right)=\sum_{t=1}^{T}\min\left(\frac{\pi_{\theta }\left(a_{t}|s_{t}\right)}{\pi_{\theta old}\left(a_{t}|s_{t}\right)}A_{t},clip\left(\frac{\pi_{\theta }\left(a_{t}|s_{t}\right)}{\pi_{\theta old}\left(a_{t}|s_{t}\right)},1-\epsilon ,1+\epsilon \right)A_{t}\right)$14:        Update Actor parameter θ by caculate gradient w.r.t. J(θ)15:        $L\left(\phi \right)=\sum_{t=1}^{T}{\left[\hat{R_{t}}-V_{\phi }\left(s_{t}\right)\right]}^{2}$16:        Update Critic parameter ϕ by caculate gradient w.r.t. L(ϕ)17:    **end for**18:    ***event.set***19:
**end for**



## 5. Experiments

The environment building parameters are shown in Table 2, the training hyperparameters are set as shown in Table 3, and the training scenarios are shown in Table 4. We set the training and testing stage frame size as n=16, rendering mapping size s=30, and rendering screen size as 480∗480. Testing found that rendering the potential field in the range of $\left[80,160\right]$ improved the algorithm performance when the mapping size increased. Still, when m was set above 160, the improvement of the algorithm effect was obviously weakened; thus, we set the potential field remapping parameter to m=160.

### 5.1. PPO Baseline Compared with Distributed Sample Extraction PPO

To compare the performance of distributed sample extraction PPO with PPO baseline in the path planning task, we compared the accumulated rewards of the training phase, as shown in Figure 10.

Analyzing the training results, the accumulated reward of the PPO baseline converged to a value of −210.5 at 230 episodes, but, as the training proceeded, the reward showed a dropping trend, which means that an attenuation occurred in policy, and this status could not improve, even with increased training. The distributed sample extraction PPO algorithm uses multiple workers to collect samples from different regions. On the one hand, this sampling method increases the number of samples; on the other hand, it ensures that the dataset samples are collected in each region in a balanced manner, hence increasing the stability of training. In the improved sampling policy, the accumulated reward from the several workers was significantly higher than that obtained by the random position generation based on the initial region in the PPO baseline.

### 5.2. Comparison of Reinforcement Learning and Traditional Path Planning Algorithms for Planning Paths

To compare the difference between RL and traditional path planning methods in the planned paths, we selected two scenarios. We set the same initial status in each system and used the A* algorithm and RL method to compare. The scenario settings and the results of the planned paths of the two methods are shown in Figure 11.

Figure 11a,b shows the paths searched by the A* and RL in two different scenarios. The top figure shows the A*-searched trajectory, and the bottom shows the RL-searched trajectory. As seen in Figure 11a,b, the trajectory that RL searched was smoother than A*, and the planned paths did not have trouble with being too close to obstacles. Due to a large number of reefs near small islands, deceleration and detours are usually performed when passing through the reef area to ensure navigation safety. The paths of A* did not take into account the problem of the safety of navigation. In contrast, routes with economies and security could be planned with RL.

### 5.3. Beta Policy and Gaussian Policy

To compare the performance of the Beta and Gaussian policies in the path planning task, we trained the Beta and Gaussian policies in two different dimension state-spaces and compared the accumulated reward of each worker, as shown in Figure 12. Meanwhile, the path planning test was driven for both state-spaces, comparing the planned path length distribution and the routed success rate in different policies, as shown in Figure 13.

Comparison of accumulated reward during training:

The Beta and Gaussian policies were trained for five rounds. Figure 12 shows the reward with the corresponding mean curves and error intervals. In the worker0 and worker3 cases, the Beta policy had a higher reward than the Gaussian policy during the training, in terms of both data sequence and image. Meanwhile, the Gaussian policy in the worker1 and worker2 cases oscillated in the late training period, i.e., the issue of policy decline occurred. This occurred because the policy certainty of the Gaussian gradually increases during the training, with the gradual decrease in σ leading to the high variance in the Gaussian, which creates a heavy oscillation when it is close to 600 episodes. In worker3, the Gaussian policy based on the data sequence and image had an obvious decay after 100 iterations. Conversely, the Beta policy had a more obvious recovery after 400 episodes, and the reward of the Beta was always greater than the Gaussian at the same iteration. Since the area corresponding to worker4 was closer to the endpoint, Gaussian and Beta demonstrated less difference in accumulative rewards;

Comparing the reward convergence of these two state-spaces in training, it was found that the number of iterations required for the image to converge in training was significantly less than that of the data sequence. This is mainly because the image provides more comprehensive global information, enabling the agent to have more feature information to be considered in execution, thus planning a more comprehensive path because the global information is knowable. However, because the state-space based on the data sequence cannot obtain complete information in the environment and can only detect through radar data, the data sequence takes a long time to acquire samples from interacting, so the image converges faster than the data sequence. In summary, the Beta policy learned by each worker significantly improved compared to Gaussian, and, in terms of stability and robustness, Beta was also higher than the Gaussian policy;

Comparison of planning path length:

Next, we took the Beta and Gaussian policy models with the same number of training iterations for the tests of data sequence and image. In testing, we used the same test position in Section 4, and each part was used for 200 iterations of path planning. We drew a box diagram of the planned length of each starting position according to the results, and compared the distance between the quartiles and the range of the extreme points by box plots to determine the stability of the planned path length. Figure 13 shows the test results. The Beta policy was significantly better than the Gaussian in the path planning stability in different state-spaces, and this advantage was shown in the case of testing at the same initial position, in which the median of Area1 was nearly 50% higher in the Beta policy (data sequence 171, image 180) than the Gaussian policy (data sequence 329, image 370). The differences between Q3 and Q1 of the Beta policy in Area2, Area4, and Area5 were lower than those of the Gaussian (more obviously in the data sequence case), and the distribution of planning path length was relatively concentrated. As well, the median of the data was significantly lower than that of the Gaussian strategy, indicating that the Beta strategy was more stable during the test;

Planning shortest path length:

According to the shortest path length analysis based on the above data, as shown in Table 5 and Figure 13, the path length of Gaussian policy search in regions 1, 4, and 5 was 4.1%, 6.5%, and 43.9% higher than that of Beta, and the path length of Gaussian policy search in regions 2, 3, and 6 was not much different from that of Beta;

Planning success rate:

As shown in Table 6, the planning success rates of Gaussian in regions 3, 4, and 5 were not as expected, but, in Beta, the success rate performance was higher in all areas, and the planning success rate was higher than 75% in each region.

### 5.4. Comparison of Distributed Sampling PPO Based on Beta Policy and Traditional Path Planning Algorithm

In this model, reinforcement learning (RL) used continuous action space exploration for global path planning. It leveraged Box2D for continuous angle manipulation and simulated ship engine propulsion effects in the action space framework. This approach resulted in smoother trajectories compared to traditional algorithms.

To verify whether there were other aspects of performance loss with smoother tracks in RL, we created experiments to compare the performance of distributed extractor PPO based on a Beta policy with the traditional algorithms. We refactored the code of the heatmap obstacles to generate the rasterized map. We discretized a map to a size of 160∗160 and marked barriers, in particular, to generate the map, which corresponded to simulations. The traditional algorithms searched the track in a rastered map and mapped the nodes searched to the dimensions of the simulation to determine the distance. The comparison between reinforcement learning path planning and traditional path planning algorithms was mainly assessed in three aspects: planning time, the number of planning loop operations, and path length.

The time required for the algorithm to perform path planning in different areas (Table 7) and the number of planning loop operations (Table 8), as well as the planned path length (Table 9), were considered separately during the test. We compared the traditional algorithms IDA*, Dijkstra, A*, and bidirectional A* with the reinforcement learning algorithm TD3 [23], as well as the optimized Gaussian policy PPO algorithm (Normal-PPO) and the Beta policy optimized PPO algorithm (Beta-PPO).

From Table 7, Table 8 and Table 9, Dijkstra took the longest time of the traditional algorithms and required the most iterations, but obtained the best solution for regions 3 and 4. IDA took significantly longer to track than other algorithms in regions 1, 3, and 5. However, its search paths were the shortest in each region, compared with different algorithms of the same type. The Bi-directional A* took significantly shorter planning time than other algorithms in each area and had the lowest number of calls among traditional algorithms; however, Bi-directional A* did not guarantee that the searched path was the best solution.

During the comparison of reinforcement learning algorithms, the training occurred under the same hyperparameters, including interaction environment parameters, batch size, and number of iterations. The best policy model from the training process was then selected for testing. Upon analyzing the data in Table 7, Table 8 and Table 9, we observe that the TD3 algorithm performs markedly worse than other algorithms for planned path length in regions 1–6. Additionally, it requires the highest number of algorithm calls among the reinforcement learning algorithms. Based on this analysis, we can conclude that the TD3 algorithm is notably less effective than the optimized PPO algorithm in generating random initial regions.

In the comparison between the optimized Gaussian policy PPO algorithm and the Beta policy optimized PPO algorithm, the Gaussian policy requires fewer algorithm calls than the Beta policy, resulting in better planning time in testing. However, the experimental results show that the Beta policy has a better planning success rate and path length than the Gaussian policy. This is because the Beta policy employs more algorithm calls in the planning process, enabling finer maneuvers during path search that lead to shorter paths in global path planning. When comparing the differences between RL and traditional path planning algorithms in terms of planning time, number of algorithm calls, and path length, as shown in Table 7, Table 8 and Table 9, we see that RL considers the propulsion characteristics of the ship for continuous control in path planning. This approach yields smoother trajectories than traditional algorithms, and RL’s planning task performance and path length are comparable to that of conventional algorithms.

## 6. Conclusions

This study aims to address the limitations of both traditional planning algorithms and reinforcement learning algorithms in the context of path planning tasks. Traditional planning algorithms require precise environment modeling, while most reinforcement learning algorithms can only perform path planning in discrete action spaces. To this end, this paper designs a generic environment framework for path planning tasks, builds the environment structure using the Box2D physics engine, and implements the environment design for continuous state-space and action space for reinforcement learning. On this basis, a deep reinforcement learning path planning, without precise modeling to enable dynamic decision-making, is realized by designing two state-spaces based on sensor data and image data.

In addition, this study solves the problems that reinforcement learning faces in path planning tasks, such as the unreasonable design of the reward function, using deterministic policies, and incomplete consideration of ship propulsion characteristics, as well as that traditional planning algorithms increase computation when facing scenario enlargement and other situations and cannot handle high-dimensional data. In this study, a reward function is designed to match the state-space, which considers the distance between the agent and the arrival point as the main part and incorporates the potential-based superposition of the arrival point, obstacles, and boundary control as the auxiliary parts. Moreover, the stochastic policy algorithm PPO is employed as a baseline for path planning. This enables the reinforcement learning algorithm to learn from the continuous angle control and engine propulsion features simulated by Box2D, effectively incorporating the ship’s propulsion characteristics into the path planning process.

Building upon this, we propose a distributed sampling PPO algorithm based on the Beta policy. This algorithm addresses the issue of sample collection imbalance in global path planning tasks and effectively improves the performance and robustness of the algorithm. By resolving the action boundary problem associated with Gaussian policy, the proposed approach enhances the algorithm’s performance in terms of accumulated reward and planning success rates across different sub-regions.

However, this study still has certain limitations. Specifically, the absence of testing in real marine environments during the experiments may lead to potential performance degradation in practical applications. To address this limitation, we are currently engaged in the deployment of the algorithm on hardware to assess the performance of the deep reinforcement learning global path planning algorithm in the context of sim2real. Furthermore, this study did not include a comparison with genetic algorithms for path planning. As another essential algorithm in path planning, genetic algorithms deserve investigation regarding their performance within the environment framework designed in this study. Thus, we will introduce genetic algorithms into the path planning environment designed in this study to evaluate their performance. 

## Figures and Tables

**Figure 1 sensors-23-06101-f001:**
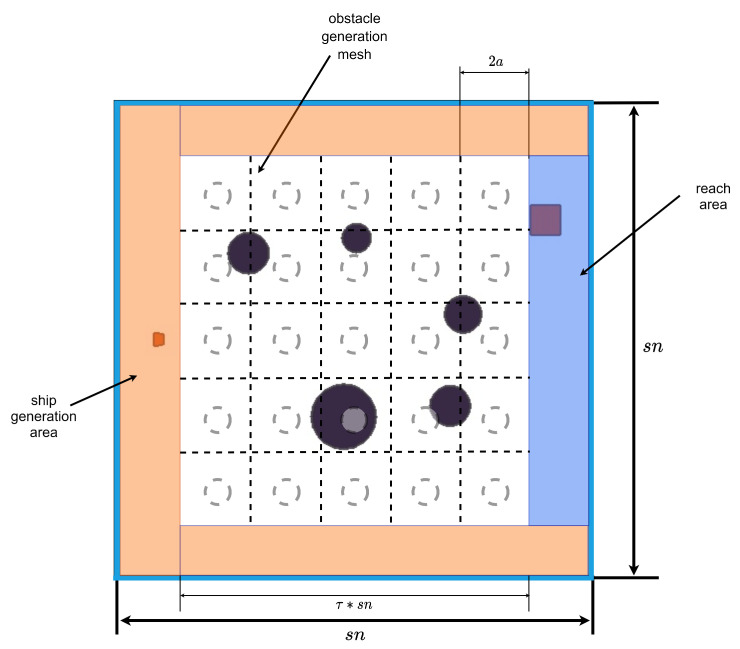
Location of simulation environment generation.

**Figure 2 sensors-23-06101-f002:**
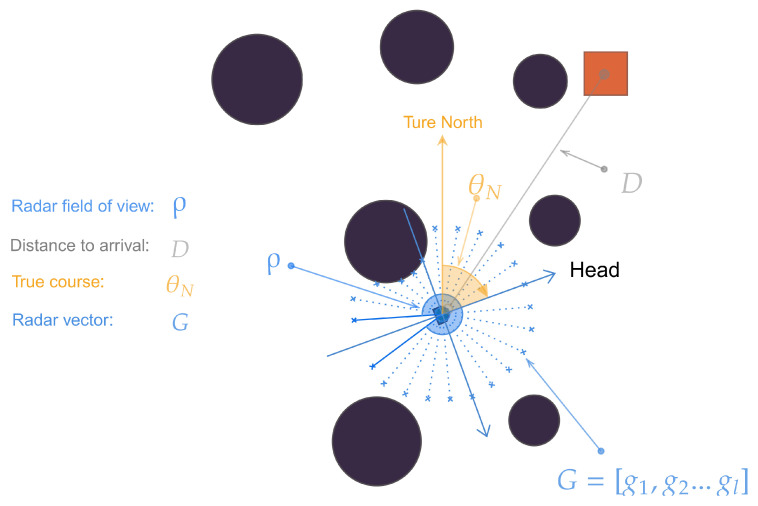
Data sequence state-space.

**Figure 3 sensors-23-06101-f003:**
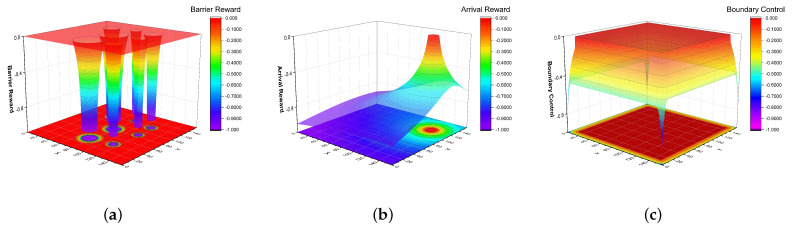
Potential-based reward: (**a**) Obstacle-based potential plane; (**b**) Arrival point-based potential plane; (**c**) Boundary control-based potential plane.

**Figure 4 sensors-23-06101-f004:**
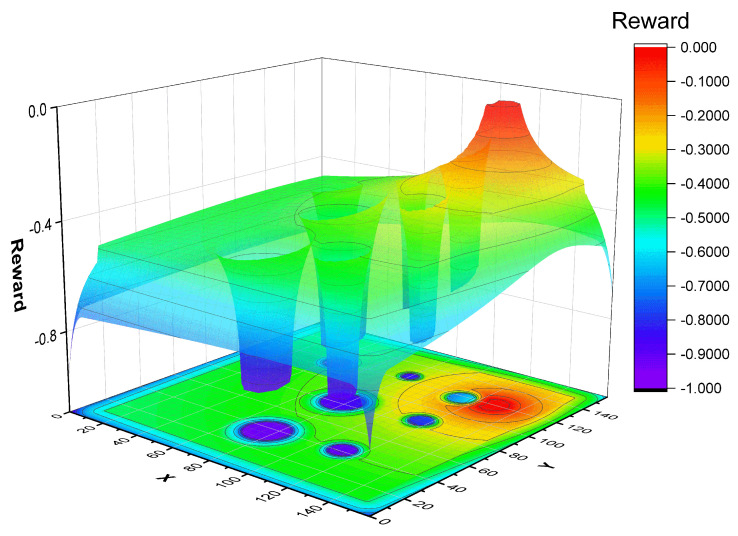
Potential plane based on obstacle, arrival point, and boundary control (superimposed).

**Figure 5 sensors-23-06101-f005:**
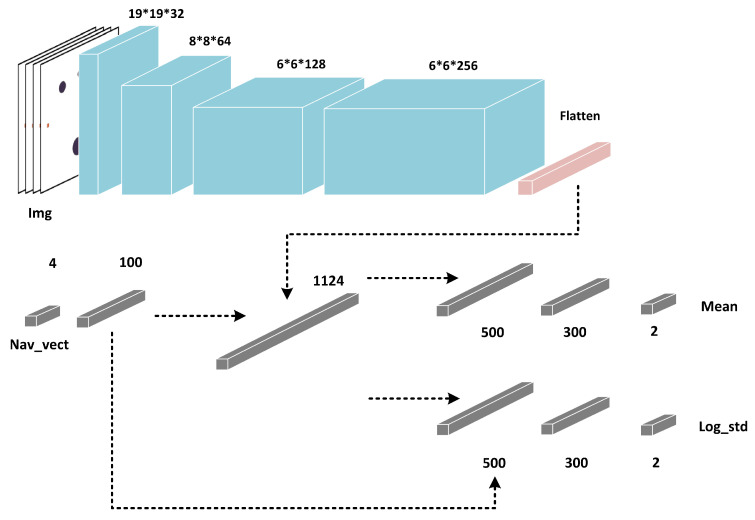
Actor network structure.

**Figure 6 sensors-23-06101-f006:**
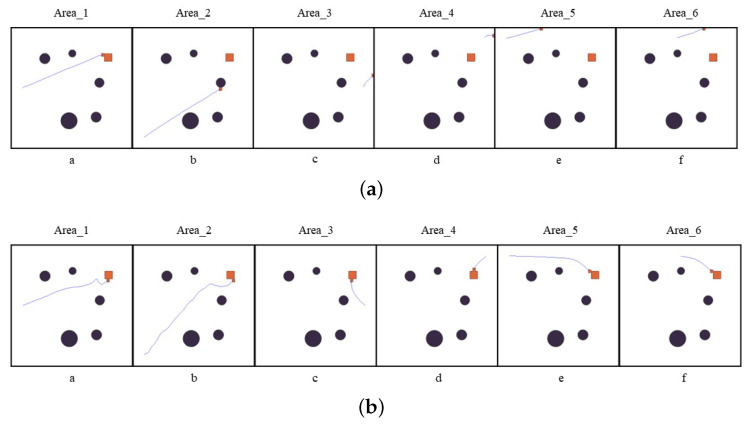
Testing results: (**a**) PPO baseline planning test; (**b**) Distributed fixed starting position test.

**Figure 7 sensors-23-06101-f007:**
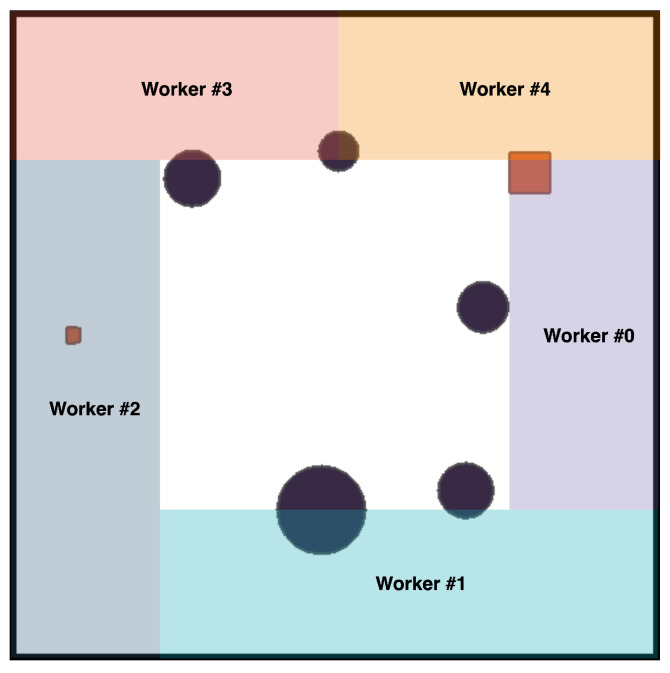
Refine sub-regions.

**Figure 8 sensors-23-06101-f008:**
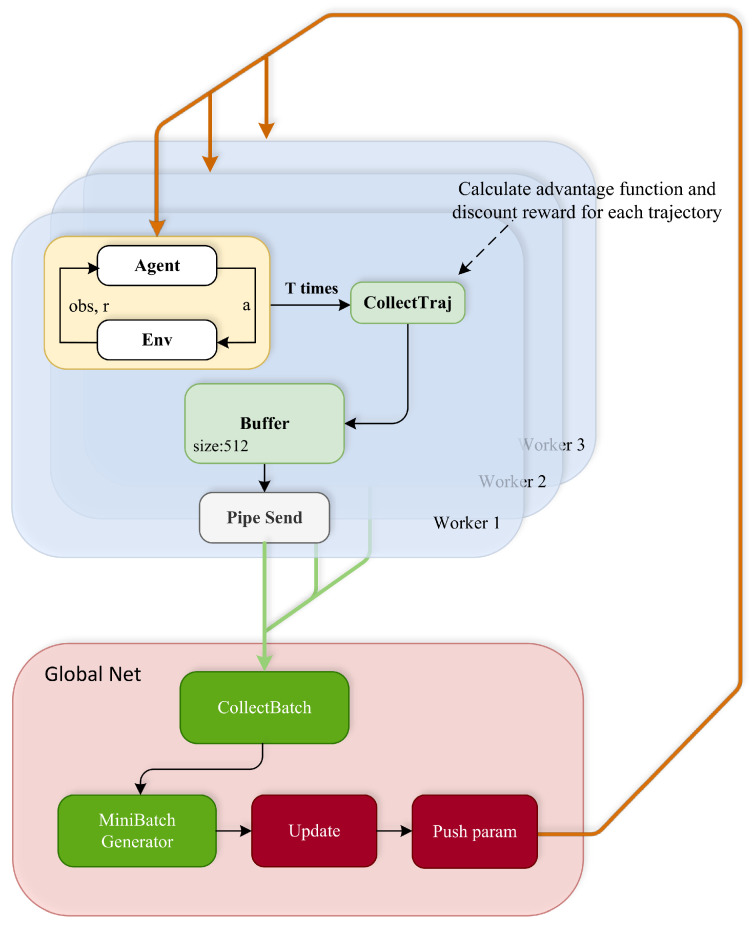
Distributed PPO algorithm.

**Figure 9 sensors-23-06101-f009:**
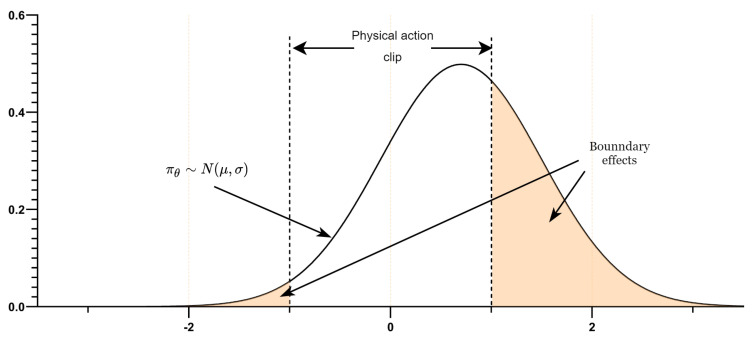
Gaussian boundary effects.

**Figure 10 sensors-23-06101-f010:**
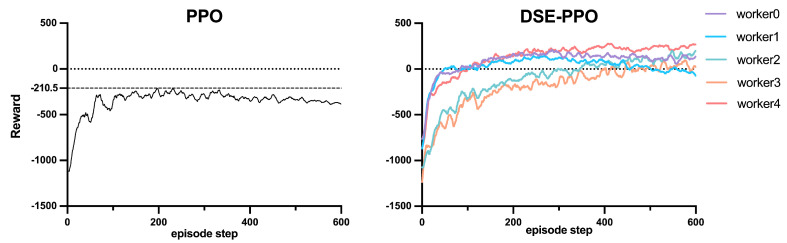
Accumulated rewards of PPO baseline compared with distributed sample extraction PPO.

**Figure 11 sensors-23-06101-f011:**
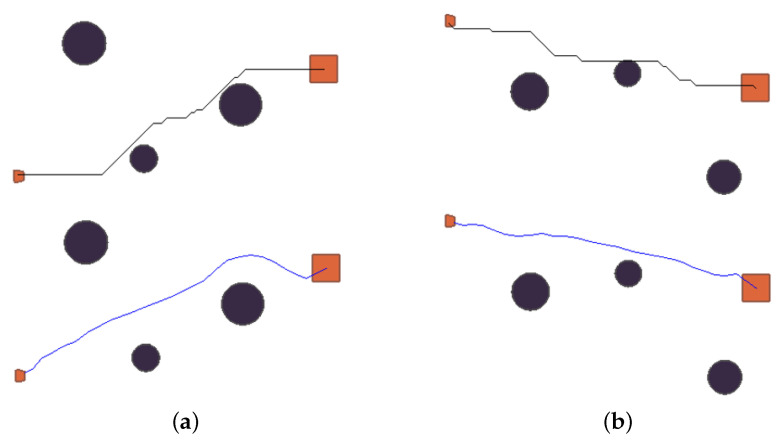
Planning path comparison: (**a**) A* and RL search path in scenario1; (**b**) A* and RL search path in scenario2.

**Figure 12 sensors-23-06101-f012:**
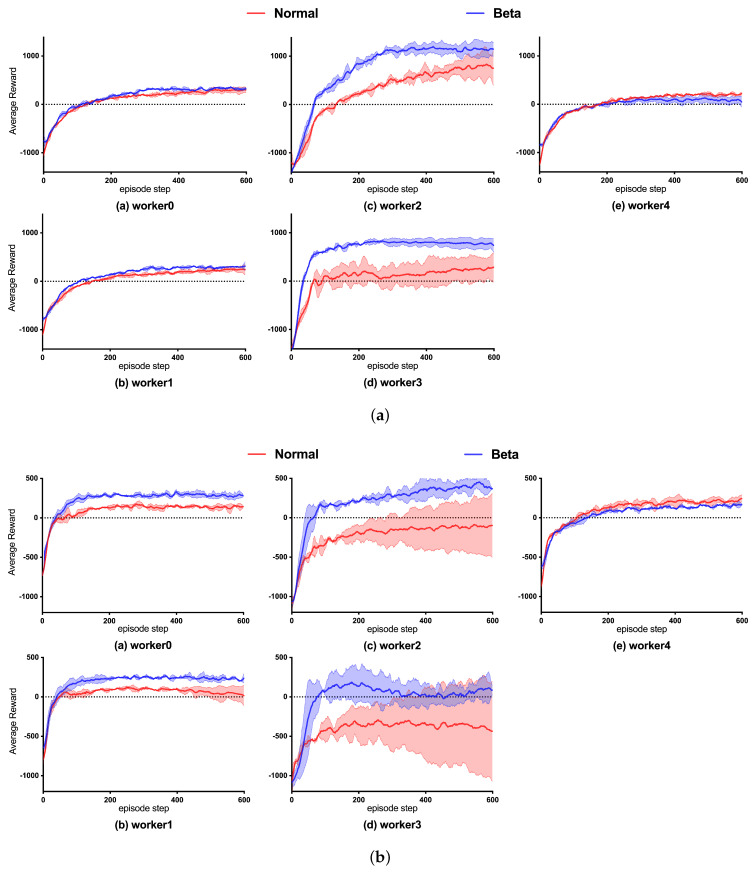
Sub-region accumulated rewards: (**a**) Data sequence state-space; (**b**) Image state-space.

**Figure 13 sensors-23-06101-f013:**
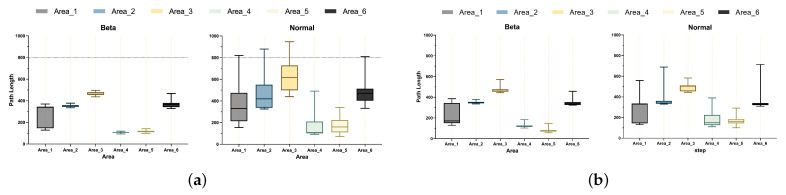
Comparison of planning path length: (**a**) Data sequence state-space; (**b**) Image state-space.

**Table 1 sensors-23-06101-t001:** Related work.

Algorithm	Deterministic/ Stochastic Policy	Action Space	Advantages	Disadvantages
Traditional Path Planning	Deterministic	Discrete	High computational efficiency; easy to adjust algorithm parameters.	Unable to directly process high-dimensional data; As the size of the scene map increases, the calculation increases; Unable to provide the recommended sailing information in path planning; Requires accurate environmental modeling; The ship kinematic model is not considered.
Sun Proposed	Deterministic	Discrete	Consider the ship’s dynamics model.	Discrete action space.
DCQN + A*	Deterministic	Discrete	Shorter convergence time by combining Q-learning and A*.	Discrete action space; Dynamic model simplification limits propulsion strategy design comprehensiveness; The reward function design lacks reasonability and overlooks the planning boundary problem.
DRLOAD	Deterministic	Discrete	Propose safe obstacle avoidance and target approach (reach point approach) reward function.	Discrete action space; Ship navigation capability is not considered in planning; The reward function design is too simple.
Kuczkowski Proposed	Deterministic	Continuous	Design the reward functions for different sub-objectives.	The ship kinematic model is not considered.

**Table 2 sensors-23-06101-t002:** Experiment environment parameters.

Parameter		Value
Plane frame structure size	*n*	16
Render scale	*s*	30
Obstacle grid size	*k*	5
Obstacle generation ratio	τ	0.6
Ship radius	*b*	0.36
Laser horizontal field angle	ρ	360
Laser effective reflection range	*h*	3.6
Laser beam	*l*	24
Reward function mapping size	*m*	160
$obsc_{R}$ based offset	α	1.5
$reach_{R}$ based offset	β	6.0
$dist_{L}$ based offset	γ	3.0

**Table 3 sensors-23-06101-t003:** Hyperparameters.

Parameter		Value
Actor training episode	$E_{\pi }$	16
Actor learning rate	$lr_{\pi }$	30
Actor learning rate decay	$Step{LR}_{\pi }$	5
Actor gradient clipping (L2)	$cn_{\pi }$	0.6
Critic training Episode	$E_{v}$	0.36
Critic training Episode	$lr_{v}$	360
Critic learning rate	$StepLR_{v}$	3.6
Critic learning rate decay	$gnorm_{v}$	24
Critic gradient clipping (L2)	γ	160
Discount factor	ϵ	1.5
Clip ratio	β	6.0
Episode training iter	$train_{iter}$	3.0
GAE lambda	λ	6.0
Adapt KL target	$KL_{tg}$	6.0
Batch Size	$batch_{iter}$	6.0
Max Step	$max_{timestep}$	6.0

**Table 4 sensors-23-06101-t004:** Training set.

Parameter	Value
CPU	Intel Xeon E5 2666v (Boost 3.5 GHz)
Memory	Quard Channel Team DDR4-2133
Motherboard	Msi X99A SLI
Strong	SAMSUNG 840 EVO 1T
GPU	2∗Gainward DuraPro GTX 1080 Ti Hurricane

**Table 5 sensors-23-06101-t005:** Shortest planning path.

Policy	Area1	Area2	Area3	Area4	Area5	Area6
Beta	**129.83**	**335.52**	**444.53**	**103.06**	**56.28**	**324.59**
Normal	135.50	338.99	444.58	110.27	100.39	327.88
Disparity	4.1%	1.0%	0.0%	6.5%	43.9%	1.0%

**Table 6 sensors-23-06101-t006:** Planning success rate.

Policy	Area1	Area2	Area3	Area4	Area5	Area6
Beta	**0.87**	**0.76**	**0.88**	**0.92**	**0.89**	**0.79**
Normal	0.85	0.76	0.31	0.36	0.15	0.71

**Table 7 sensors-23-06101-t007:** Algorithm planning time comparison (in seconds).

Policy	Area1	Area2	Area3	Area4	Area5	Area6
IDA*	13.799	0.051	86.633	0.119	9.287	0.051
Dijkstra	1.371	5.015	5.302	0.79	0.237	3.111
A*	0.057	7.118	2.231	0.057	0.037	1.961
Bi-A*	0.083	0.509	0.975	0.058	0.048	0.182
TD3	0.046	0.102	0.134	0.038	0.024	0.095
Beta	0.038	0.074	0.093	0.029	0.029	0.071
Normal	**0.025**	**0.049**	**0.072**	**0.029**	**0.022**	**0.047**

**Table 8 sensors-23-06101-t008:** Comparison of planning operands (iterative loop operands).

Policy	Area1	Area2	Area3	Area4	Area5	Area6
IDA*	1.02 × 10^6^	385	6.58 × 10^6^	9.14 × 10^3^	7.09 × 10^5^	359
Dijkstra	6.47 × 10^3^	1.96 × 10^4^	2.07 × 10^4^	4.12 × 10^3^	1.85 × 10^3^	1.44 × 10^4^
A*	433	7.82 × 10^3^	3.77 × 10^3^	262	236	2.31 × 10^3^
Bi-A*	452	1.85 × 10^3^	2.31 × 10^3^	364	382	923
TD3	15	36	45	12	8	33
Beta	10	21	26	8	7	20
Normal	**6**	**12**	**17**	**7**	**6**	**13**

**Table 9 sensors-23-06101-t009:** Planning path length comparison (∗0.1 km).

Policy	Area1	Area2	Area3	Area4	Area5	Area6
IDA*	134.26	349.66	428.67	97.32	**55.32**	333.50
Dijkstra	137.27	349.66	**427.43**	**96.08**	58.32	333.51
A*	140.5	353.26	434.76	99.31	59.53	335.43
Bi-A*	141.76	355.39	440.11	105.05	61.42	333.15
TD3	159.29	369.10	467.07	132.75	87.45	351.83
Beta	**129.83**	**335.52**	444.53	103.06	56.28	**324.59**
Normal	135.50	338.99	444.58	110.27	100.39	327.88

## Data Availability

Not applicable.

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
