# Peer review of "An Improved Distributed Sampling PPO Algorithm Based on Beta Policy for Continuous Global Path Planning Scheme"

_sensors, 2023, doi:10.3390/s23136101_

Round 1

Reviewer 1 Report

The technical details about the work is good but the beta policy can be explained more.

The ideology is good and the implementation part diagrams and results are in convincing range.

The language is clear and good.

Author Response

Dear reviewer, thank you very much for providing the valuable comments and suggestions. We highly appreciate your evaluation. We have carefully considered each of your comments and made revisions and improvements to the article. Also, we have replied to each of your comments and hope that you are satisfied with our responses. Please see the attachment. Thank you again for your effort and support.

Reviewer 2 Report

This paper proposes a  path planning algorithm based on improved distributed sampling Proximal Policy Optimization algorithm.  The main topic discussed in this paper is path planning, which is not highly related to sensors. It may fail to capture the readers' interest of Sensors. This paper is not suitable for publication in Sensors and The manuscript may be transferred to a robotic journal.

other comments:

1. The path planning of single point robot have been discussed for several decades. There are GA based approaches which might be more suitable than ML-based approaches. The authors might compare these two approaches.

2. The proposed algorithm is a potential based approach. It is important to discuss how to avoid the local minimum. 

Author Response

(The authors gave the same response as above.)

Reviewer 3 Report

The paper has presented state-of-the-art PPO (Proximal Policy Optimization) algorithm as a baseline for global path planning to address the issue of incomplete ship navigation power propulsion strategy. To improve the performance of global path planning algorithms, deep reinforcement learning is increasingly used to solve global path planning problems. It shows an interesting paper. However, some questions in this paper are as follows:

1-    What are technical contributions in this paper?  It should be sated in the introduction section.

2-    The proposed model/ system architecture should be drawn and explained in this paper.

3-    What are weaken points of your methods? It should be scoped in the conclusion section.

The paper should be rewritten in the conclusion section to state results, main points, limitations, and future works grouped in paragraphs.

No comments.

Author Response

(The authors gave the same response as above.)

Reviewer 4 Report

The paper might include valuable contribution, but it is difficult to evaluate that with confidence. It contains a lot of information, but the presentation seems a bit chaotic. Therefore, it is advised to enhance the clarity of presented ideas and results, considering the following detailed remarks:

1. It is advised to list related works, mentioned in Section 1 (Introduction) in a table, comparing their most important features, such as: global or local, deterministic or stochastic, advantages, limitations, optimization criteria, etc. On the basis of this comparative analysis it is advised to list the limitations of existing methods, the approach presented in this paper eliminates. 

2. It is advised to add flowcharts of the proposed PPO algorithm.

3. All of the abbreviations should be resolved in the first sentence they occur, e.g. DQN, DDPG.

4. All of the parameters in the equations given in the paper should be explained e.g. parameters used in Equation (2.1).

5. In Fig.11 it is advised to compare A* and RL path using one figure (a and b in one figure, c and d in another figure) + enlarge the figures

6. It is advised to update Fig. 14. – enlarge and change, so that it would be easier to compare the paths calculated by different algorithms, as now the differences are not readily apparent.

7. Please give the units of path length used in Table 8. 

Please also try to rewrite the paper a little bit in order to make it easily comprehensible, as some sentences are very long and therefore difficult to understand, e.g. on page 2 “The limitations of these path planning algorithms include the following: first, these algorithms require the system to be known when used for global path planning, which results in that if the environment changes, such as when new obstacles appear, it is necessary to re-acquire accurate info about the environment and re-plan the route. “. Some sentences are written as an instruction, e.g. on page 4 lines 141-147, it is advised to rephrase such parts of the text.

Author Response

(The authors gave the same response as above.)

Round 2

Reviewer 3 Report

The paper has been addressed the reviewer comments. However,  Line 300,  Consistent with the angle treatment, 299 the propulsion power is also normalized and mapped to interval [−1, 1]. Please correct it why not [1,1] ? explain it.

It should be checked English language carefully for improvement of the paper.

Author Response

(The authors gave the same response as above.)

Reviewer 4 Report

The paper has been improved significantly and in the current form is suitable for publication in the Sensors journal.

Author Response

Thank you very much for your affirmation and support of our work. Your valuable suggestions and opinions have played a crucial role in our research, and we are deeply grateful for that. Once again, thank you for your support of our work. We will continue to strive for excellence to deliver better results.